Deciphering the influence of evolutionary legacy and functional constraints on the patella: an example in modern rhinoceroses amongst perissodactyls

http://orcid.org/0000-0002-1982-3803 Mallet Christophe 1 2 cmallet@naturalsciences.be
Houssaye Alexandra 3
1 Faculty of Engineering, University of Mons, Department of Geology and Applied Geology , Mons , Belgium
2 Institute of Natural Sciences, Operational Directorate Earth and History of Life , Brussels , Belgium
3 Muséum National d’Histoire Naturelle, Mécanismes adaptatifs et évolution (MECADEV), UMR 7179, MNHN, CNRS , Paris , France
Abdala Virginia
Electronic publication date: 2024 Oct 25
Publication date: 2024
Volume: 12
Electronic Location ID: e18067
Received 2024 Feb 29; Accepted 2024 Aug 19
Copyright: © 2024 Mallet and Houssaye
Copyright year: 2024
Copyright holder: Mallet and Houssaye
License: This is an open access article distributed under the terms of the Creative Commons Attribution License, which permits unrestricted use, distribution, reproduction and adaptation in any medium and for any purpose provided that it is properly attributed. For attribution, the original author(s), title, publication source (PeerJ) and either DOI or URL of the article must be cited.
License URL: https://creativecommons.org/licenses/by/4.0/

Keywords: Functional morphology, Rhinocerotidae, Equidae, Tapiridae, 3D geometric morphometrics, Sesamoid bone, Allometry, Knee joint

Funding: European Research Council and is part of the GRAVIBONE project ERC-2016-STG-715300 This work was funded by the European Research Council and is part of the GRAVIBONE project (ERC-2016-STG-715300). The funders had no role in study design, data collection and analysis, decision to publish, or preparation of the manuscript.

==============================
In mammals, the patella is the biggest sesamoid bone of the skeleton and is of crucial importance in posture and locomotion, ensuring the role of a pulley for leg extensors while protecting and stabilizing the knee joint. Despite its central biomechanical role, the relation between the shape of the patella and functional factors, such as body mass or locomotor habit, in the light of evolutionary legacy are poorly known. Here, we propose a morphofunctional investigation of the shape variation of the patella among modern rhinoceroses and more generally among perissodactyls, this order of ungulates displaying a broad range of body plan, body mass and locomotor habits, to understand how the shape of this sesamoid bone varies between species and relatively to these functional factors. Our investigation, relying on three dimensional geometric morphometrics and comparative analyses, reveals that, within Rhinocerotidae and between the three perissodactyl families, the shape of the patella strongly follows the phylogenetic affinities rather than variations in body mass. The patellar shape is more conservative than initially expected both within and between rhinoceroses, equids and tapirs. The development of a medial angle, engendering a strong mediolateral asymmetry of the patella, appears convergent in rhinoceroses and equids, while tapirs retain a symmetric bone close to the plesiomorphic condition of the order. This asymmetric patella is likely associated with the presence of a “knee locking” mechanism in both equids and rhinos. The emergence of this condition may be related to a shared locomotor habit (transverse gallop) in both groups. Our investigation underlines unexcepted evolutionary constraints on the shape of a sesamoid bone usually considered as mostly driven by functional factors.

Introduction

In vertebrates, the shape of the skeleton is strongly influenced by structural, functional, developmental constraints, and evolutionary legacy (Seilacher, 1970, 1991; Gould, 2002). Limb bones, which function in like body support and locomotion in quadrupeds, are particularly influenced by functional constraints (Hildebrand, 1974; Hall, 2007). Variations of body plan, body mass or locomotor habits induce changes in the shape of limb elements (Hildebrand, 1974; Polly, 2007; Biewener & Patek, 2018), as observed in many clades of tetrapods (see for example Fabre et al., 2013; MacLaren & Nauwelaerts, 2016; Etienne et al., 2020a; Serio, Raia & Meloro, 2020). However, despite this intense exploration of form-function relationships focused mainly on limb long bones, smaller elements like sesamoid bones remain poorly studied. Sesamoids are small concentrations of bone generally embedded in dense connective tissue such as tendons or ligaments. They are assumed to increase leverage for muscles and tendons attached to them, while protecting the joints with which they are associated (Vickaryous & Olson, 2008). The emergence of sesamoids in some joints is directly linked to variations in mechanical loads, making them likely to carry a strong functional signal (Vickaryous & Olson, 2008; Eyal et al., 2019).

The patella is the largest sesamoid bone of the skeleton (Samuels, Regnault & Hutchinson, 2017). Long considered to form within the distal quadriceps tendon, recent findings show that the patella arises as a bony process at the anterodistal surface of the femur (Eyal et al., 2015, 2019). The patella is later embedded into the quadriceps tendon and articulates with the femur by sliding onto the distal trochlear groove. Onto the patella attach the different ends of the musculus quadriceps as well as the m. gluteobiceps, all being knee extensors (Barone, 2010a; Samuels, Regnault & Hutchinson, 2017; Etienne, Houssaye & Hutchinson, 2021) (Fig. 1). The patella is also connected to the tibia through the patellar tendon, and additional lateral and medial tendons (Barone, 2010a; Samuels, Regnault & Hutchinson, 2017). The main functional role of the patella is to modify the lever arm of the m. quadriceps. By moving the quadriceps tendon away from the centre of rotation of the knee joint, the patella acts as a pulley and reduces the muscular energy required to extend the leg, while increasing the velocity of the rotation (Aglietti & Menchetti, 1995; Allen et al., 2017). In mammals, the patella also centralizes the forces coming from the four heads of the m. quadriceps into a single point of application and transmits them without friction to the tibial tuberosity. In addition, the patella acts as a protective element for the knee joint (Aglietti & Menchetti, 1995).

Figure 1 Main muscular insertions on the patella of a white rhinoceros (Ceratotherium simum).

Muscle insertions replaced on the 3D model of Ceratotherium simum cottoni AMNH M-51854 based on the anatomical descriptions of Barone (2010a) and Etienne, Houssaye & Hutchinson (2021).

The functional importance of the patella led to various works that explored the biomechanical involvement of this sesamoid bone in the hindlimb movements and its mechanical advantage, mostly in humans and related hominoids (Bizarro, 1921; Ellis et al., 1980; Aglietti & Menchetti, 1995; Lovejoy, 2007; Dan et al., 2018; Pina et al., 2020; Schneider, Rooks & Besier, 2022). These aspects have been rarely addressed in other groups of amniotes (Alexander & Dimery, 1985; Chadwick et al., 2014; Allen et al., 2017), most of the studies focusing more on the evolutionary relevance of the presence or absence of the patella (Jerez, Mangione & Abdala, 2010; Ponssa, Goldberg & Abdala, 2010; Corina Vera, Laura Ponssa & Abdala, 2015; Abdala, Vera & Ponssa, 2017; Samuels, Regnault & Hutchinson, 2017). While the inner bony structure of the patella has sometimes been explored in regards to functional constraints (Raux et al., 1975; Toumi et al., 2006; Houssaye, de Perthuis & Houée, 2021), the variation of its external shape, considered as relatively conservative in amniotes (Samuels, Regnault & Hutchinson, 2017), remains largely understudied. Only a few studies tried to investigate the form-function relationships at a larger interspecific scale, mostly in mammals (Valois, 1917; Raymond & Prothero, 2012; Pina et al., 2020; Garnoeva, 2022). Therefore, the variation of the shape of the patella in relation with functional constraints remains virtually unknown in amniotes.

In this context, we chose to explore the shape variation of the patella in modern rhinoceroses and compare it with shape variation within perissodactyls (adding equids and tapirs) to investigate its relations with functional factors within a phylogenetically informed framework. Modern rhinoceroses constitute the second heaviest land mammal group after elephants (Alexander & Pond, 1992), and contrary to them, show great variations (factor of 6) of body mass (BM) among the five living species: from 600 kg for Dicerorhinus sumatrensis (Fischer, 1814) up to 3,500 kg for Ceratotherium simum (Burchell, 1817). These three-toed animals inhabit various environments from open savannas to dense rainforests, associated with diverse feeding strategies, from specialized grazers to generalist browsers (Dinerstein, 2011). They perform transverse gallop (Economou et al., 2020) and are able to reach speeds from 27 km.h−1 for Ceratotherium simum to 45 km.h−1 for Diceros bicornis (Garland, 1983; Alexander & Pond, 1992; Blanco, Gambini & Fariña, 2003). Body mass in rhinos can vary strongly between males and females in some species (Zschokke & Baur, 2002; Dinerstein, 2011), as well as their body plan, from long-legged black rhino Diceros bicornis to short-legged white rhino C. simum. This diversity of body construction and ecological preferences makes them a particularly well-suited group for studying the link between patellar shape and functional factors.

To expand the scope of our investigation, we chose to include in a second time other modern perissodactyls as comparative groups. The other modern perissodactyls comprise seven species of equids and four of tapirs. These two clades show less intragroup variations of size and body plan. Tapirs are forest-dwellers with a limited range of body mass (factor of 2): from 175 to 400 kg (Medici, 2011). Tapirs retain some plesiomorphic characters of the order Perissodactyla in their limbs, such as a tetradactyl manus and a tridactyl pes (MacLaren & Nauwelaerts, 2016, 2017). Their running pace is different from that of rhinos, as they perform a rotary gallop (Economou et al., 2020). As for equids, they show marked weigh variations related to domestication, although not comparable to what is observed among rhinos. If most wild equids weigh around 300 kg, some domestic horses and donkeys (such as draft races) can exceed 800 kg (factor of 2.6). In addition, equids show a highly derived limb structure adapted for running and living in open habitats, with monodactyl manus and pes (Rubenstein, 2011), associated with a transverse gallop as for rhinos (Economou et al., 2020). These three families (Rhinocerotidae, Tapiridae, Equidae) also show a gradient of stature and proportions in their limb construction, from cursorial equids to “mediportal” tapirs and “graviportal” rhinos, likely to be associated with differences in patellar shape (Eisenmann & Guérin, 1984). Recent advances in functional morphology underlined how body mass could drive–or not–the shape of limb bones in perissodactyls (MacLaren & Nauwelaerts, 2016, 2017; Hanot et al., 2017, 2018; MacLaren et al., 2018; Mallet et al., 2019, 2020, 2021, 2022; Etienne et al., 2020b), but without consideration for the patella.

Moreover, rhinoceroses and equids are known to possess a “knee locking” mechanism in their hindlimb: they are able to lock the patella and the medial patellar tendon above the medial trochlear ridge of the distal femur (Hermanson & MacFadden, 1996; Schuurman, Kersten & Weijs, 2003). Doing so, the natural flexion of the limb that could be engendered in relation with body weight becomes impossible, allowing these animals to save muscular energy for long periods of standing. Long considered as passive, in vivo experiments demonstrated that this mechanism does require muscular energy, even though far much less than would be needed without this apparatus (Schuurman, Kersten & Weijs, 2003). Broadly studied in domestic equids for veterinary purposes, this locking mechanism is less documented in vivo in rhinos and only inferred through the shape of the distal femur but not directly through that of the patella (Shockey et al., 2008; Danaher, Shockey & Mihlbachler, 2009; Janis et al., 2012; Mihlbachler et al., 2014). A recent investigation on the microanatomy of the patella in modern perissodactyls highlighted microanatomical changes related to different functional factors, notably the presence of the knee locking mechanism (Houssaye, de Perthuis & Houée, 2021). But a similar comparative investigation remains to be conducted for deciphering how functional factors are related to the external shape of the patella in this group.

For all these reasons, we chose to explore both intra- and interspecific variation of the shape of the patella among modern rhinoceroses, and to compare it with tapirs’ and equids’, to shed light on how this crucial sesamoid bone is influenced by functional constraints (body mass, locomotor habit, presence or not of a “knee locking” mechanism) and evolutionary legacy among Perissodactyla. We relied on a 3D geometric morphometrics approach to investigate the form-function relationships on the patella. Given previous results on the shape variation of hindlimb long bones (Mallet et al., 2019, 2020) and on the inner structure of the patella (Houssaye, de Perthuis & Houée, 2021) in modern rhinos, we hypothesize: (1) higher interspecific than intraspecific variation of the shape of the patella in rhinos; (2) a patellar morphology not explained by phylogenetic relations alone but also by functional factors such as body mass, presence or absence of a knee locking mechanism, and type of gallop in perissodactyls (Schuurman, Kersten & Weijs, 2003; Shockey et al., 2008; Economou et al., 2020). The consideration of equids and tapirs will allow us to situate the variation observed among rhinoceroses within the perissodactyls as a whole, and see whether the trends observed, and the hypotheses of functional links highlighted in rhinoceroses are confirmed by this contextualization on the scale of the whole family, or whether different trends between families could call into question the suggested links between form and function.

Materials and Methods

Studied sample

We selected a sample composed of 54 patellae of modern perissodactyls housed in eight institutions (American Museum of Natural History, New York, USA; Powell Cotton Museum, Birchington-on-Sea, UK; Idaho Museum of Natural History, Pocatello, USA; Muséum National d’Histoire Naturelle, Paris, France; Museum of Vertebrate Zoology, Berkeley, USA; Natural History Museum, London, UK; Naturhistorisches Museum Basel, Basel, Switzerland; Institute of Natural Sciences, Brussels, Belgium) and representing the five modern species of rhinos (27 patellae), together with four modern species of tapirs (12 patellae) and seven species (nine subspecies) of modern equids (15 patellae) (Table 1, Fig. 2). We used binomial names as provided in the Handbook of Mammals of the World (vol. 2) (Dinerstein, 2011; Medici, 2011; Rubenstein, 2011). The sample includes 29 males, 17 females and 19 specimens without sex attribution. It involves only adult specimens (e.g., having fully fused epiphysis on the long bones associated with the patella), except for one specimen of unclear age but showing a general adult aspect and size and one Equus zebra hartmannae considered as subadult (e.g., having not fully fused epiphysis on the associated long bones) (see Table 1 for details). This last one has been kept in the sample as it was the only available specimen we encountered for this subspecies. Because of the higher intrafamilial diversity of rhinos, we chose to focus first on this group. We performed our analyses successively on rhinoceroses only, then on rhinos, equids, and tapirs together. Exploration of shape variation among equids and tapirs respectively are provided as Supplemental Data. All anatomical terms follow classic anatomical and veterinary works (Federative Committee on Anatomical Terminology, 1998; Barone, 2010b), and are given in Fig. 3.

Table 1 List of the studied specimens with family, genus and species names, mean body mass, institutions, sex, age class, condition, and 3D acquisition details.

Family	Taxon	Mean body mass (kg)	Institution	Collection number	Sex	Age	Condition	3D acquisition	
Rhinocerotidae	Ceratotherium simum	2,300	AMNH	M-51854	F	A	W	SL	
			AMNH	M-51855	M	A	W	SL	
			AMNH	M-51857	F	A	W	SL	
			AMNH	M-51858	M	A	W	SL	
			AMNH	M-81815	U	A	U	SL	
			BICPC	NH.CON.37	F	A	W	SL	
			BICPC	NH.CON.110	M	A	W	SL	
			BICPC	NH.CON.112	M	A	W	SL	
			MNHN	ZM-MO-2005-297	M	A	C	CT	
			NHMUK	ZD 2018.143	U	A	U	SL	
	Diceros bicornis	1,050	AMNH	M-113777	U	A	W	SL	
			MNHN	ZM-AC-1944-278	M	A	C	CT	
			NHMUK	ZD 1879.9.26.6	U	U	U	CT	
	Dicerorhinus sumatrensis	775	AMNH	M-81892	M	A	W	SL	
			NHMUK	ZD 1879.6.14.2	M	A	W	SL	
			NHMUK	ZD 1894.9.24.1	U	A	W	SL	
			NHMUK	ZE 1948.12.20.1	U	A	U	SL	
			NHMUK	ZE 1949.1.11.1	U	A	W	SL	
			NHMUK	ZD 2004.23	U	A	W	SL	
	Rhinoceros sondaicus	1,350	MNHN	ZM-AC-A7970	U	A	U	SL	
			NHMUK	ZD 1871.12.29.7	M	A	W	SL	
	Rhinoceros unicornis	2,000	AMNH	M-35759	M	A	C	SL	
			AMNH	M-54454	F	A	W	SL	
			MNHN	ZM-AC-1967-101	F	A	C	SL	
			NHMUK	ZE 1961.5.10.1	M	A	W	SL	
			NHMUK	ZD 1972.822	U	A	U	SL	
			NHMUK	ZD 1884.12.1.2	F	A	W	SL	
Tapiridae	Tapirus bairdii	260	MVZ	141172	U	A	U	LS*	
	Tapirus indicus	340	MNHN	ZM-AC-1931-528	M	A	C	CT	
			MNHN	ZM-AC-1945-460	M	A	C	CT	
			NMB	8125	F	A	C	CT	
			RBINS	1184D	M	A	C	LS	
			RBINS	1184E	M	A	C	LS	
	Tapirus pinchaque	175	MNHN	ZM-AC-1982-34	M	A	C	CT	
			MNHN	ZM-AC-1877-765	U	A	U	CT	
	Tapirus terrestris	220	MNHN	ZM-AC-1937-1	U	A	C	CT	
			MNHN	ZM-MO-1990-20	M	A	C	CT	
			RBINS	1185D	M	A	C	LS	
			RBINS	1185E	U	A	C	LS	
Equidae	Equus africanus asinus	275	MNHN	ZM-AC-1893-634	M	A	C	CT	
			MNHN	ZM-2005-717	U	A	U	CT	
			RBINS	12970	F	A	C	LS	
			RBINS	13076	F	A	C	LS	
	Equus hemionus	230	MNHN	ZM-AC-1880-1103	M	A	C	CT	
	Equus ferus caballus	490	MNHN	ZM-AC-A541	F	A	C	CT	
			MVZ	162289	M	A	C	LS*	
	Equus ferus przewalskii	250	MNHN	ZM-AC-1975-124	F	A	C	CT	
			RBINS	14281	M	A	C	LS	
	Equus burchellii granti	247	RBINS	33386	U	A	U	LS	
	Equus grevyi	400	RBINS	32166	M	A	C	LS	
	Equus quagga boehmi	247	RBINS	12129	M	A	W	LS	
	Equus quagga chapmani	247	RBINS	1218	F	A	C	LS	
	Equus quagga quagga	247	IMNH	R2425	U	A	U	LS*	
	Equus zebra hartmannae	310	RBINS	3974	M	S	C	LS	
Note:

Abbreviations: Sex–F, female; M, male; U, unknown. Age–A, adult; S, sub-adult. Condition–W, wild; C, captive; U, unknown. 3D acquisition–LS, laser scanner; SL, structured light surface scanner; P, photogrammetry; CT, CTscan. Stars indicate specimens retrieved on MorphoSource deposit. Institutional codes–AMNH, American Museum of Natural History. BICPC, Powell Cotton Museum, Birchington-on-Sea. IMNH, Idaho Museum of Natural History, Pocatello. MNHN, Muséum National d’Histoire Naturelle, Paris. MVZ, Museum of Vertebrate Zoology, Berkeley. NHMUK, Natural History Museum, London. NMB, Naturhistorisches Museum Basel, Basel. RBINS, Royal Belgian Institute of Natural Sciences, Brussels.

Figure 2 Composite cladogram of the sampled species.

Branch lengths and relations based on Steiner & Ryder (2011), MacLaren et al. (2018), Bai et al. (2020), Cirilli et al. (2021), Antoine et al. (2021), Liu et al. (2021), Pandolfi et al. (2021). Silhouettes of C. simum, Dc. bicornis, Ds. sumatrensis, E. z. hartmannae, E. grevyi, R. sondaicus, R. unicornis and T. indicus are personal creations. All other silhouettes provided by www.phylopic.org under the Creative Commons license.

Figure 3 Location of the landmarks used for the analyses.

A total of six anatomical landmarks (red spheres), 103 curve sliding (blue spheres) and 461 surface sliding (green spheres) semi-landmarks were placed on the patella. Designation of the anatomical landmarks–1, Most proximal point of the medial ridge; 2, most distal point of the medial ridge; 3, most medial point of the medial articular surface; 4, most proximal point of the medial articular surface; 5, most proximal point of the base; 6, most medial point of the medial angle.

3D models

Twenty-four patellae were digitized using a structured-light three dimensional scanner (Artec Eva) and reconstructed with Artec Studio professional (v12.1.1.12; Artec 3D, Luxembourg, Belgium). Twelve specimens were scanned using a Creaform HandySCAN 300 laser surface scanner and reconstructed using the VX Models software (Creaform, Houston, TX, USA). Fifteen patellae were scanned using high-resolution computed tomography at the micro-CT laboratory of the Natural History Museum, London, UK (Nikon HMX 225 ST system), and six at the AST-RX platform at the Muséum National d’Histoire Naturelle, Paris (UMS 2700; GE phoenix|X-ray v|tome|xs 240), with reconstructions performed using the CT-agent software (Nikon Metrology, Leuven, Belgium), and DATOX/RES software (phoenix datos|x). Bone surfaces were extracted as meshes using Avizo 9.4 (https://www.thermofisher.com/fr/fr/home/electron-microscopy/products/software-em-3d-vis/avizo-software.html; Thermo Fisher Scientific, Waltham, MA, USA). Three specimens were retrieved on the online deposit MorphoSource (see Table 1 for details). Only right bones were selected for digitization; when unavailable, we selected left bones instead and mirrored the 3D models before analysis using MeshLab (v2022.02; Cignoni et al., 2008). ARK identifiers of each 3D model are provided in Table ST1.

3D geometric morphometrics

We analysed the shape variation of our sample using a 3D geometric morphometrics approach. This widely used methodology allows to quantify and visualize the morphological differences between objects by comparing the spatial coordinates of landmarks placed on them (Adams, Rohlf & Slice, 2004; Zelditch et al., 2012; Mitteroecker & Schaefer, 2022). The shape of the patella was quantified by placing a set of anatomical landmarks and curve and surface sliding semi-landmarks on the meshes, as described by Gunz, Mitteroecker & Bookstein (2005) and Botton-Divet et al. (2016). Anatomical landmarks and curves were placed on meshes using the IDAV Landmark software (v3.0–Wiley et al., 2005). We defined six anatomical landmarks associated with two curves allowing to cover the shape diversity of the patella in modern perissodactyls (Fig. 3). We evaluated the relevance of these anatomical landmarks to describe the shape by conducting repeatability tests. We considered three specimens of Ds. sumatrensis chosen to display the closest morphology. We successively digitized ten times the anatomical landmarks on the three specimens to obtain thirty replications of the landmark datasets. We superimposed these landmark configurations using a generalized Procrustes analysis (GPA, see below) then performed a principal component analysis (PCA) to visualise the placement of each repeated set of measurement relatively to each other. It appeared clearly that the variation within specimens was lower than between specimens, confirming the relevance of our landmarks to precisely describe the shape variation (see Fig. S1).

Anatomical landmarks and curve sliding semi-landmarks were placed on each specimen. A template was created for the placement of the surface sliding semi-landmark on each patella. A specimen (C. simum AMNH M-51854) was selected as the initial specimen on which all anatomical landmarks, curve and surface sliding semi-landmarks were placed. This specimen was selected for its average shape and size allowing to ensure the correct placement of the landmarks on all the other specimens despite the great variation of size and shape in the sample. It was then used as a template for the projection of surface sliding semi-landmarks on the surface of all other specimens. Projection was followed by a relaxation step ensuring that projected points matched the surface of the meshes. Curve and surface semi-landmarks were then slid by minimizing the bending energy of a thin plate spline (TPS) between each specimen and the template at first, then four times between the result of the previous step and the Procrustes consensus of the complete dataset (see Gunz, Mitteroecker & Bookstein, 2005; Botton-Divet et al., 2016 for details regarding this process). Once this process was complete, all landmarks could be treated as geometrically homologous. We then performed a GPA to remove the effect of relative size, location and orientation of the different landmark conformations (Gower, 1975; Rohlf & Slice, 1990). Projection, relaxation, sliding processes and GPA were conducted using the “Morpho” package (v2.10; Schlager, 2017) in the R environment (v4.2.1; R Core Team, 2014). Details of the function parameters used are provided in the R code available as Supplemental Data.

Superimposed conformations of landmarks were used to compute a PCA in order to reduce dimensionality (Gunz & Mitteroecker, 2013) (function “procSym” in the “Morpho” package). PC scores where then used to compute neighbour-joining (NJ) trees and morphospaces allowing to visualise the distribution of each individual relatively to each other. To visualise how shape varies in association with the specimen distribution, we computed theoretical shapes associated with both minimal and maximal values for the two first PCs using a TPS deformation of a template mesh. To emphasize the location of the main shape changes on the patella among the sample, we displayed the vectors of displacement between all landmarks of the two theoretical shapes using the “rgl” package (v0.109.6; Adler & Murdoch, 2020). As described by Botton-Divet (2017) and Pintore et al. (2021), we applied a gradient of colour to these segments according to their distance to highlight which parts varied the most, these segments being displayed directly onto the superimposed theoretical shapes (see the R code available as Supplemental Data for details of the process). Detailed NJ trees and PCA plots for equids and tapirs only are provided as Supplemental Data.

We explored the relation between shape and specific attribution, sex, size, type of gallop and presence/absence of a knee-locking mechanism (see below). Permutational analyses of variance (PERMANOVAs) (Costa-Pereira et al., 2016; Anderson, 2017) were performed on PC scores against the specific attribution and sex class to test for the impact of these parameters on the patellar shape. We also performed a PERMANOVA on PC scores against the type of gallop (transverse and rotary) to address the effect of locomotor mode. As the presence of a knee-locking mechanism is distributed exactly as the type of gallop (i.e., tapirs have a rotary gallop and no knee-locking mechanism, while equids and rhinos have a transverse gallop and a knee-locking mechanism), we did not test for this last parameter, as the result would have been redundant with the one previously described. We used the function “adonis2” in the “vegan” package (v.2.6-4) to perform PERMANOVAs (using the Euclidian method and removing lacking sex attribution with the argument “na.exclude”) (Dixon, 2003). We explored allometric variation (i.e., shape variation linked to size; Hallgrímsson et al., 2009; Zelditch et al., 2012; Mitteroecker et al., 2013; Klingenberg, 2016) within our sample using different approaches. Pearson’s correlation tests were performed to test for correlation between the scores of the two first PCs and the centroid size of the specimens. We performed a Procrustes ANOVA (a linear regression model using Procrustes distances between species instead of covariance matrices; Goodall, 1991; Adams & Otárola-Castillo, 2013) with 1,000 permutations and a sequential sum of squares to quantify the shape variation related to the centroid size to highlight the influence of centroid size on the global shape variation. Procrustes ANOVA has been performed using the function “procD.lm” in the “geomorph” package. This set of analyses (PERMANOVA, Pearson’s correlation tests, Procrustes ANOVAs) was applied respectively on rhinoceroses alone and on all perissodactyls.

Proxies of body mass

In the absence of most of the mass values of the specimens constituting our sample, we chose two proxies of body mass: (1) the centroid size (CS) of each patella, generally used to address allometric variation, defined as the square root of the sum of the square of the distance of each point to the centroid of the landmark set (Zelditch et al., 2012); it is generally considered as a good proxy of the body mass of the animal; and (2) the minimal femoral circumference (FC) of each specimen, a measurement classically used in equations of body mass estimation. Indeed, the femur being a crucial bone for weight support in quadrupeds, its circumference is known to strongly covary with body mass (Anderson, Hall-Martin & Russell, 1985; Scott, 1990; Damuth & MacFadden, 1990; Campione & Evans, 2012; Campione, 2017). FC therefore provides a proxy of mass relatively independent of the patella itself. We used a Pearson’s correlation test to evaluate the correlation between CS and FC. We also performed Procrustes ANOVAs as described above using FC instead of CS to quantify their respective relations with shape variation, both on rhinos alone and on all perissodactyls.

Phylogenetic framework

To test the presence of a phylogenetic signal in our shape data, we constructed a composite cladogram using trees previously computed on molecular data and craniodental and postcranial characters (Fig. 2). Branch relations, lengths and occurrence dates were retrieved and merged following phylogenetic reconstructions proposed in recent works (Steiner & Ryder, 2011; MacLaren et al., 2018; Bai et al., 2020; Cirilli et al., 2021; Antoine et al., 2021; Liu et al., 2021; Pandolfi et al., 2021). The phylogenetic relation of Ds. sumatrensis relatively to the other modern species has long been debated (see Mallet et al., 2019). As recent works based on molecular and morphological data agreed on the phylogenetic position of this species (Antoine et al., 2021; Liu et al., 2021; Pandolfi et al., 2021), we considered here Ds. sumatrensis as sister taxon of the Rhinoceros genus.

We addressed the effect of phylogenetic signal on shape and centroid size in our dataset. Given the small numbers of rhino species (five), tests for phylogenetic signal were only performed on all perissodactyl species. For species represented by several specimens, we computed species mean shapes (Botton-Divet et al., 2017; Serio, Raia & Meloro, 2020; Mallet et al., 2021, 2022). We computed a first GPA with all specimens of the second subsample (all perissodactyls) as described previously, then we computed the Procrustes consensus (or mean shape) of each species in this geometric space. These Procrustes consensuses were superimposed in a second GPA before computing a new PCA. We then addressed the effect of phylogeny on mean centroid sizes per species with the univariate K statistic (Blomberg et al., 2003), using the function “phylosig” in the “phytools” package (Revell, 2012). We also addressed the effect of phylogenetic relationships on shape data using the multivariate K statistic (Kmult) on PC scores (Adams, 2014). Kmult index allows the comparison between the rate of observed morphological change and that expected under a Brownian motion on a given phylogeny (Blomberg et al., 2003; Adams, 2014). Kmult was computed using the function “physignal” in the “geomorph” package (v4.0.4–Adams & Otárola-Castillo, 2013).

According to recent works calling for a continuous approach of the p value (Wasserstein, Schirm & Lazar, 2019; Ho et al., 2019), statistic tests were considered as significant for p values ≤ 0.05 but we chose to mention and comment results having a p value between 0.05 and 0.10 as well.

Results

Shape variation within rhinoceroses

The distribution of the specimens both in the NJ tree and in the morphospace shows a clear distinction between African and Asiatic rhinos. Along the NJ tree (Fig. 4), specimens of C. simum and Dc. bicornis plot together and oppose to Ds. sumatrensis, R. sondaicus and R. unicornis. The Dc. bicornis and C. simum specimens are mixed, as are the R. sondaicus and R. unicornis specimens. Those of Ds. sumatrensis form an isolated and more homogeneous cluster. The morphospace of the first two axes of the PCA, representing 47.5%, displays a similar structure (Fig. 5A). PC1, which carries 33.6% of the global variance, separates African rhinos towards negative values from Asiatic rhinos towards positive values. C. simum, occupies the highest negative values of the axis while R. unicornis and R. sondaicus occupy the highest positive values. Ds. sumatrensis specimens partially overlap R. unicornis along the PC1. PC2 represents 13.9% of the variance and is structured by the isolation of Ds. sumatrensis towards the highest positive values, while all other species overlap around lower values.

Figure 4 Neighbour-joining tree computed on all PC scores obtained from the PCA performed on shape data of rhinoceroses.

Colour code follows Fig. 2. Letters indicate sex attribution as in Table 1 (F, female; M, male; U, unknown). Point size is proportional to the log centroid size of each specimen. Silhouettes of C. simum, Dc. bicornis, Ds. sumatrensis, E. z. hartmannae, E. grevyi, R. sondaicus, R. unicornis and T. indicus are personal creations. All other silhouettes provided by www.phylopic.org under the Creative Commons license. Tree generated by our R code provided as Supplemental Data.

Figure 5 Results of the PCA performed on morphometric data of rhinoceroses and associated shape variation.

(A) Morphospace of the two first axes of the PCA with minimal and maximal theoretical shape associated with this variation (respectively in medial and caudal views). Colour codes follow Fig. 2. Letters indicate sex attribution as in Table 1. Point size is proportional to the log centroid size of each specimen. Silhouettes of C. simum, Dc. bicornis, Ds. sumatrensis, E. z. hartmannae, E. grevyi, R. sondaicus, R. unicornis and T. indicus are personal creations. All other silhouettes provided by www.phylopic.org under the Creative Commons license. (B) Morphological variation between minimal (light brown) and maximal (light grey) theoretical shapes along a) PC1 and b) PC2 respectively in medial, cranial, lateral, caudal, dorsal (top) and ventral (bottom) views. Intensities of landmark displacements are shown with vector colorations ranging from blue (low distance) to red (high distance). Plot and theoretical 3D models generated by our R code provided as Supplemental Data (using the specimen Dicerorhinus sumatrensis AMNH M-81892 as a template for deformation of the meshes).

A Pearson’s correlation test indicates that PC1 (r = −0.41; p = 0.03) and PC2 (r = −0.55; p < 0.01) are correlated with the centroid size (Table 2). The PERMANOVA of PC scores against the specific attribution and sex of the specimen confirms a highly significant correlation between the shape of the patella and the specific attribution (R² = 0.59; p < 0.01), but no correlation with the sex attribution (p = 0.47) (Table 3).

Table 2 Results of the Pearson’s correlation tests between the log-transformed centroid size and the two first principal components of the PCA, for rhinoceroses only and all perissodactyls respectively.

	Component	r	t	dF	P	
Rhinoceroses	PC1	−0.41	−2.24	25	0.03	
	PC2	−0.55	−3.278	25	<0.01	
All perissodactyls	PC1	−0.61	−5.60	52	<0.01	
	PC2	−0.48	−3.96	52	<0.01	
Note:

Abbreviations: r, Pearson’s correlation coefficient value; t, Student distribution value; dF, degrees of freedom; P, p value. Significant results are indicated in bold.

Table 3 Results of the PERMANOVAs performed on PC scores against specific attribution and sex, for rhinoceroses only and all perissodactyls respectively, and on PC scores against type of gallop for all perissodactyls.

		Df	Sum of squares	R²	F	P	
Rhinoceroses	Species	4	0.08	0.59	4.20	<0.01	
	Sex	1	<0.01	0.03	0.90	0.47	
	Residuals	11	0.05	0.38			
	Total	16	0.13	1			
All perissodactyls–Sex	Species	15	0.62	0.86	8.34	<0.01	
	Sex	1	<0.01	<0.01	0.74	0.563	
	Residuals	20	0.10	0.14			
	Total	36	0.72	1			
All perissodactyls–Type of gallop	Gallop	1	0.43	0.40	35.36	<0.01	
	Residuals	52	0.63	0.60			
	Total	53	1.05	1			
Note:

Abbreviations: dF, degrees of freedom; R², coefficient of partial determination; F, F-statistics; P, p value. Significant results are indicated in bold.

Shape variation along PC1 is mainly related to the proximodistal extension of the base and the aspect of the medial angle (Fig. 5B-a). Towards minimal values, patella display a rounded and smooth general aspect. The articular surface for the femur is concave and has a convex cranial surface. The proximal end of the medial articular surface slightly exceeds that of the lateral articular surface. The medial ridge is sigmoid in medial view, with a rounded distal part. The base is mediolaterally broad and extends slightly proximally, being only little higher than the proximal border of the articular surface. The medial angle is rounded and smooth, pointing mediodistally. Towards maximal values, the patella has a more angular general aspect. The cranial surface of the patella forms a straight dorsoventral line, before displaying a caudoventral inflexion towards the apex. The proximal end of the medial articular surface exceeds greatly that of the lateral articular surface. The medial ridge is straight in medial view, forming an almost flat relief caudally to the apex. The base forms a saliant triangle with a narrow base, and a right angle markedly higher than the proximal border of the articular surface. The medial angle forms a rounded protrusion pointing strictly medially.

Shape variation along PC2 is mainly dominated by changes in the antero-posterior broadness of the patella, the proximodistal extension of the base and the apex, and the aspect of the medial angle (Fig. 5B-b). Towards minimal values, the patella has a robust general aspect, especially antero-posteriorly, with a base extending strongly proximally. The articular surface for the femur has a medial part pointing in proximal direction. In medial view, the anterior surface of the patella follows a straight line from the base before forming a marked obtuse angle towards the apex. The proximal end of the medial articular surface exceeds markedly that of the lateral articular surface. The medial ridge is slightly concave, almost straight, in medial view. The base is thick, broadly triangular and forms a saliant proximal protrusion. The medial angle forms a broad triangle pointing directly medially. The apex is thick and short. Towards maximal values, the patella is flatter antero-posteriorly, with a prominent protrusion of the apex. The articular surface for the femur has an irregular triangular shape pointing distally. The anterior surface of the patella forms a convex surface from the base to the apex. The proximal end of the medial articular surface slightly exceeds that of the lateral articular surface. The medial ridge is concave in medial view and forms a convex relief above the apex. The base is very short and slightly extended proximally. The medial angle forms a large, rounded protrusion pointing medially. The apex expands largely and forms a saliant relief directed posterodistally.

The regression plot of the Procrustes ANOVA performed on shape against log centroid size shows a marked difference between Asiatic and African rhinos (Fig. 6A). Towards maximal CS values, C. simum and Dc. bicornis plot above the line while most R. unicornis plot below the line. Towards minimal CS values, most Ds. sumatrensis plot below the line or close to it, except for one individual showing a lower centroid size compared to other specimens. Positions of R. sondaicus relatively to the line are hard to interpret, given the small number of individuals and the isolation of an unusually small specimen towards minimal CS values. Shape variation associated with centroid size mainly involves changes in the general proportions of the patella, as well as the relative development of the base, the medial angle and, at a lesser extent, the apex (Figs. 6A and 6B). Towards high values of centroid size, the patella broadens antero-posteriorly, showing a more rounded and smooth general aspect. The base is broad and thick, slightly developed proximally, forming a convex relief posteriorly to the articular surface. The medial angle forms a smooth triangle pointing medially, distally, and posteriorly. The apex appears short and broad.

Figure 6 Results of the Procrustes ANOVA on shape data against log-transformed centroid size (CS) for rhinos.

(A) Regression plot with theoretical shapes associated with minimum and maximum fitted values (respectively in medial and caudal views). Colour code follows Fig. 2. Point size is proportional to the log centroid size of each specimen. Symbols indicate the sex attribution (triangle: male; circle: female; square: unknown). Silhouettes of C. simum, Dc. bicornis, Ds. sumatrensis, E. z. hartmannae, E. grevyi, R. sondaicus, R. unicornis and T. indicus are personal creations. All other silhouettes provided by www.phylopic.org under the Creative Commons license. (B) Colour maps of the location and intensity of the shape deformation. The shape associated with the maximal CS value of the Procrustes ANOVA was coloured depending on its distance to the shape associated with the minimal value. Green indicates no deformation; blue indicates a negative deformation of high intensity; red indicates a positive deformation of high intensity. Plot and theoretical 3D models generated by our R code provided as Supplemental Data. Plot and theoretical 3D models generated by our R code provided as Supplemental Data (using the specimen Dicerorhinus sumatrensis AMNH M-81892 as a template for deformation of the meshes).

Moreover, the shape is significantly correlated with centroid size (p < 0.01) but centroid size accounts for only 12% of the global shape variance (R² = 0.12) (Table 4) The regression plot highlights a signal strongly driven by Ds. sumatrensis and one specimen of R. unicornis. When removing these two species from the Procrustes ANOVA, the correlation between shape data and size is not significant anymore (p = 0.18). However, the small number of individuals for these species prevents us to go further on this question. In accordance with the strong correlation between centroid size and femoral circumference (Pearson’s correlation test; r = 0.75, p < 0.01), very similar results are obtained when considering the log of the minimal femoral circumference instead of the centroid size in the Procrustes ANOVA (Table 4). The regression plot of shape data against FC led to an almost similar distribution of the specimens than that obtained with CS, African rhinos plotting above the regression line while the two Rhinoceros species plot below and Ds. sumatrensis isolates towards minimal FC values (see Fig. S2A). Shape variation associated with FC are very similar to that observed with CS too, with an even more marked modification of the medial ridge of the patella (see Fig. S2B).

Table 4 Results of the Procrustes ANOVAs on shape data against log centroid size and femoral circumference respectively, for rhinoceroses only and all perissodactyls respectively.

		dF	Sum of squares	R²	F	Z	P	
Rhinoceroses only	Log centroid size	1	0.03	0.12	3.37	2.57	<0.001	
	Residuals	25	0.19	0.88				
	Total	26	0.22					
	Log femoral circumference	1	0.02	0.12	3.20	2.33	0.01	
	Residuals	24	0.18	0.88				
	Total	25	0.21					
All perissodactyls	Log centroid size	1	0.23	0.22	14.74	4.13	<0.01	
	Residuals	52	0.81	0.78				
	Total	53	1.04					
	Log femoral circumference	1	0.25	0.26	17.27	4.40	<0.01	
	Residuals	50	0.73	0.74				
	Total	51	0.99					
Note:

Abbreviations: dF, degrees of freedom; R², coefficient of determination; F, F-statistics; Z, Z score; P, p value. Significant results are indicated in bold.

Shape variation between all perissodactyls

When considering all perissodactyls, shape data carries a significant but low phylogenetic signal (Kmult = 0.141, p < 0.01), while the mean centroid size per species does not carry a significant phylogenetic signal (p = 0.57). The distribution of the specimens both in the NJ tree and in the morphospace strictly separates rhinos, equids, and tapirs (Fig. 7). Among rhinos, the repartition previously observed is conserved, with clustering African and Asiatic rhinos, and one Dc. bicornis clustering close to C. simum. The species clustering is far less clear among tapirs, although the low number of specimens limits the scope of our observations. All specimens of Tapirus indicus, the largest and only Asiatic species of tapir, cluster together, while the other three species are less clustered. When considered separately, the different tapir species discriminate well on the PCA but stay unclustered on the NJ tree (see Fig. S3). Among equids, most E. quagga plot together, as do most donkeys (E. africanus asinus, E. hemionus), far from E. quagga. Between them, E. grevyi plot together with horses (E. ferus caballus and E. ferus przewalskii). Two larger specimens of E. asinus are mixed among horses and quaggas, respectively. When considered alone, equids show a strong mixing, despite most zebras and horses grouping together, respectively (see Fig. S4).

Figure 7 Neighbour Joining tree computed on all PC scores obtained from the PCA performed on shape data of all perissodactyls.

Colour code follows Fig. 2. Symbols indicate age class as in Table 1 (triangle: subadult; circle: adult). Point size is proportional to the log centroid size of each specimen. Silhouettes of C. simum, Dc. bicornis, Ds. sumatrensis, E. z. hartmannae, E. grevyi, R. sondaicus, R. unicornis and T. indicus are personal creations. All other silhouettes provided by www.phylopic.org under the Creative Commons license. Tree generated by our R code provided as Supplemental Data.

The morphospace of the first two axes of the PCA, representing 65.8% of the global variance, is structured by a strict separation between the three perissodactyl families, with overlap between the species within each family (Fig. 8A). PC1 carries 42.7% of the global variance and separates rhinos and equids in the negative part of the axis from tapirs which are positioned in the positive part. All equids group together with no clear structure, except for E. hemionus, plotting outside the equid cluster (an observation confirmed on the PCA plot of equids alone; see Fig. S4). Among each family, the separation between species is far less clear. Rhinos, to the exclusion of Ds. sumatrensis and C. simum, although to a lesser extent, have PC1 scores that overlap those of Equidae. Among tapirs, T. pinchaque isolates slightly from other species, which can also be observed on the PCA plot of tapirs alone (Fig. S3). Along PC2, which accounts for 23% of the variance, rhinos, and tapirs in the negative part of the axis are separated from equids in the positive part.

Figure 8 Results of the PCA performed on morphometric data of all perissodactyls and associated shape variation.

(A) Morphospace of the two first axes of the PCA with minimal and maximal theoretical shape associated with this variation (respectively in medial and caudal views). Colour codes follow Fig. 2. Point size is proportional to the log centroid size of each specimen. Silhouettes of C. simum, Dc. bicornis, Ds. sumatrensis, E. z. hartmannae, E. grevyi, R. sondaicus, R. unicornis and T. indicus are personal creations. All other silhouettes provided by www.phylopic.org under the Creative Commons license. (B) Morphological variation between minimal (light brown) and maximal (light grey) theoretical shapes along a) PC1 and b) PC2 respectively in medial, cranial, lateral, caudal, dorsal (top) and ventral (bottom) views. Intensities of landmark displacements are shown with vector colorations ranging from blue (low distance) to red (high distance). Plot and theoretical 3D models generated by our R code provided as Supplemental Data (using the specimen Diceros bicornis NHMUK ZD 1879.9.26.6 as a template for deformation of the meshes).

PC1 and PC2 are both highly correlated with the centroid size (Pearson’s correlation test: r = −0.61; p < 0.01 and r = −0.48; p < 0.01, respectively) (Table 2). PERMANOVA confirms an extremely high and significant correlation between the shape of the patella and the specific attribution (R2 = 0.86; p < 0.01), as well as a significant although lower correlation between shape and type of gallop (R² = 0.40; p < 0.01). Conversely, sex attribution is not significantly correlated with shape data (p = 0.56) (Table 3).

Shape variation along PC1 is mainly related to the degree of asymmetry of the patella, the extension of the medial angle, and the anteroposterior broadening of the base (Fig. 8B-a). Both extreme shapes reflect respectively rhino and tapir morphotypes. Towards negative values, the anterior surface is relatively flat, with a straight anterior border forming a well-marked angle (around 125°) towards the apex. The articular surface for the femur is globally rounded and smooth. The medial ridge is sigmoid and slightly concave in medial view. The base is broad mediolaterally and extends slightly proximally. The medial angle is rounded and smooth and extends strongly mediodistally. The apex is broad and massive antero-posteriorly, and slightly expands distally. Towards positive values, the theoretical shape of the patella is strongly symmetric, with no medial angle. The anterior surface is strongly and curved from the base to the apex. The articular surface for the femur is globally rectangular with symmetric medial and lateral facets. The medial ridge is concave in medial view, the rounded distal part towards the apex protruding only slightly. The base is well-developed and broad antero-posteriorly, ending in a marked knob pointing medio-proximally. The medial angle is almost absent, reduced to a smooth relief along the medial border of the articular surface. The apex is thin, wedge-shaped, slightly extended distally relatively to the articular surface.

Along PC2, shape variation is mainly related to the development and aspect of the base and of the medial angle (Fig. 8B-b). Towards minimal values, the shape displays a “rhino-like” general aspect with a marked mediolateral compression. The base forms a salient relief extending markedly proximally. The medial angle is rounded, smooth and extends in medial direction. Towards maximal values, patellae are shaped like a triangular-based pyramid. The anterior surface is strongly convex, extending markedly anteriorly. In medial view, the anterior border of the patella forms a 45-degrees downward slope towards the apex. The articular surface for the femur forms a marked angle pointing medially. The proximal end of the medial articular surface is situated below that of the lateral articular surface. The medial ridge forms a smooth sigmoid in medial view, the proximal part overhanging the articular surface in posterior direction. The base expands massively in anterior direction and forms a large flat surface strongly inclined distally towards the medial angle. No marked relief is visible on the surface of the base. The medial angle is broad and rounded, extending strongly medio-distally. The apex is short antero-posteriorly. Views of patellar mean shapes for each species of the dataset are displayed in the Fig. S5.

Shape is significantly correlated with centroid size (Procrustes ANOVA: R² = 0.22; p < 0.01) (Table 4). The regression plot of shape against log centroid size shows a limited spreading of the specimens around the regression line (Fig. 9A). Towards maximal CS values, C. simum and Ds. sumatrensis are now mostly above the regression line while R. unicornis plot mostly below. Once again, the position of the few specimens of Dc. bicornis and R. sondaicus remains hard to interpret. Almost all equids plot above the regression line with few exceptions: E. caballus and two E. asinus, one of which being a draught breed with the highest CS value of the sample. All tapirs plot together below the regression line and are ordered from the least body mass to the greatest one along the centroid size values. Shape variation associated with centroid size mainly reflects differences between tapirs for minimal values and rhinos for maximal values. Changes in centroid size among perissodactyls involve changes in the general asymmetry of the patella, the relative development of the medial angle and the cranio-caudal broadening of the bone (Figs. 9A and 9B). As for rhinos alone, the femoral circumference among all perissodactyls is highly correlated to the centroid size of the patella (r = 0.88, p < 0.01). Consequently, shape is also correlated with femoral circumference (Procrustes ANOVA: R² = 0.26; p < 0.01) (Table 4), with a regression plot and associated shape variation very similar to those observed for CS (see Fig. S6).

Figure 9 Results of the Procrustes ANOVA on shape data against log-transformed centroid size (CS) for all perissodactyls.

(A) Regression plot with theoretical shapes associated with minimum and maximum fitted values (respectively in medial and caudal views). Colour code follows Fig. 2. Point size is proportional to the log centroid size of each specimen. Silhouettes of C. simum, Dc. bicornis, Ds. sumatrensis, E. z. hartmannae, E. grevyi, R. sondaicus, R. unicornis and T. indicus are personal creations. All other silhouettes provided by www.phylopic.org under the Creative Commons license. (B) Colour maps of the location and intensity of the shape deformation. The shape associated with the minimal CS value of the Procrustes ANOVA was coloured depending on its distance to the shape associated with the maximal value. Green indicates no deformation; blue indicates a negative deformation of high intensity; red indicates a positive deformation of high intensity. Plot and theoretical 3D models generated by our R code provided as Supplemental Data (using the specimen Diceros bicornis NHMUK ZD 1879.9.26.6 as a template for deformation of the meshes).

Discussion

Intra- and interfamily variation of shape

Our results highlight a marked distinction between African and Asiatic rhino clades, despite important weight variations within these two groups, suggesting that the shape of the patella among rhinoceroses is highly related to phylogenetic affinities prior to body mass. Each rhino species possesses a unique patellar shape. This supports our first hypothesis that interspecific variation of patellar shape is stronger than intraspecific variation of patellar shape. In the hindlimb, the patella articulates directly with the trochlear groove of the femur, while it is linked with the tibia by only ligaments. Moreover, the patella in mammals is known to separate from the femur in the early developmental stages of the embryo (Eyal et al., 2015, 2019). The patella and the femur are therefore strongly bonded bones, both on functional and developmental aspects. These observations are congruent with previous results suggesting that femoral shape variation is also more greatly driven by phylogenetic affinities over body mass, especially the distal part directly linked with the patella (Mallet et al., 2019, 2022). This observation is however less clear for Ds. sumatrensis, as this rhino is the lightest modern species and situated outside the Rhinoceros crown-group.

Although statistically limited, our results allow one to draw preliminary considerations on the relation between the shape of the patella and sex attribution among modern rhinoceroses. Our current data do not detect a clear shape distinction between males and females. The appendicular skeleton of modern rhinos is weakly dimorphic (Guérin, 1980; Mallet et al., 2019) with phenotypic dimorphic traits being generally limited to larger horns or slightly heavier males or females (depending on the species) (Dinerstein, 1991, 2011; Zschokke & Baur, 2002). As seen above, body mass variation does not strongly influence the patellar shape in rhinos. Our results suggest that the patella does not bear a significant dimorphic signal among modern rhinos.

Among perissodactyls, the three families (Tapiridae, Equidae, Rhinocerotidae) are distinct, tapirs being very different from both rhinoceroses and equids. This result does not reflect the phylogenetic relationships among the family Perissodactyla, rhinos and tapirs being sister taxa, but rather rely on the shared high development of the medial angle, creating a strong mediolateral asymmetry, absent in tapirs. The patella condition in tapirs is plesiomorphic and thus very close to the one observed in early perissodactyls, although this bone is barely described in literature (Bai et al., 2017), while rhinos and equids show a more derived condition relatively to the supposed basal shape.

Furthermore, intrafamily variations reveal to be higher among rhinos than among equids and tapirs. These relative differences in variation within the three families may be related to their respective evolutionary history. Divergences between the different rhino clades occurred from the Oligocene to Miocene (Fig. 2) (Bai et al., 2020; Antoine et al., 2021; Liu et al., 2021). Tapir branches also diverged during the Miocene (Steiner & Ryder, 2011; MacLaren et al., 2018) but this clade remained monogeneric, contrary to rhinos. Equids, another monogeneric clade, includes species that diverged very recently during the Pliocene (Steiner & Ryder, 2011; Cirilli et al., 2021). The deeper ancestry of crown Rhinocerotidae, compared to that of crown Tapiridae and crown Equidae, may explain the higher diversity of patellar shape in this clade relatively to its sister groups.

Microanatomical investigations on the patella of modern perissodactyls revealed no increase in compactness with body mass and no thickening of the cortex on muscle insertions but on the patellar ligament on the cranial side of the bone (Houssaye, de Perthuis & Houée, 2021). These results are consistent with ours on the rather conservative nature of this bone in relation to body mass variation. Conversely, the microanatomy distinguishes more rhinos and equids than rhinos and tapirs, contrary to our results, underlining a decoupling between the external morphology of the patella and its internal structure among perissodactyls.

Functional constraints and “knee locking” mechanism

In modern rhinoceroses, our results underline a significant relation between the patellar shape and both its size and the femoral circumference, an excellent proxy of body mass in quadrupeds (Scott, 1990; Campione & Evans, 2012; Campione, 2017). The centroid size of the patella therefore appears directly related to body weight in rhinos, as previously observed for long bones and ankle bones in these animals (Mallet et al., 2019; Etienne et al., 2020b). However, shape relation with size is limited, allometry accounting for only 12% of the observed shape variation. The influence of body mass on the patellar shape therefore appears as secondary relatively to phylogenetic relationships in rhinos, which partially contradicts our second hypothesis, postulating a stronger effect of mass over phylogeny on the patellar shape. Thus, despite the central functional role of the patella in the knee, and the high stress linked to body mass exerted on this joint, the influence of the weight is much more limited than expected.

Variation linked to size mainly affects the base, the medial angle, and the lateral edge of the patella, as well as the cranial surface. All these areas are insertions for knee extensors, respectively the m. rectus femoris, the m. vastus medialis, the m. vastus lateralis (all three, together with the m. vastus intermedius, forming the m. quadriceps) and the m. gluteobiceps (Etienne, Houssaye & Hutchinson, 2021) (Fig. 1). The flattening of the base and the general craniocaudal broadening of the patella in heavy rhino species likely provide more surface area for the insertion of stronger m. rectus femoris and m. gluteobiceps, while in lighter rhinos the proximal development of the base seems to increase the attachment surface for the m. vastus lateralis and m. vastus medialis instead. This configuration may reflect that heavy rhinos rely more on central hip muscles, such as the m. rectus femoris, to stabilize the knee while lighter rhinos rely on medial and caudal muscles of the m. quadriceps, maybe ensuring more manoeuvrability for this joint. Unfortunately, the scarcity of myological data for lighter rhino species (Etienne, Houssaye & Hutchinson, 2021) and the lack of in vivo observations prevent us from making further inference between the patellar shape and its relation to attached muscles.

The allometric relation is more strongly marked when considering all perissodactyls (Fig. 9), allometry accounting for 23% of the observed variation. However, body mass alone seems insufficient to explain the shape variation among perissodactyls. A developed medial angle exists both in equids and rhinos but not in tapirs, even though tapirs and equids are in the same range of body mass (hundreds of kilograms), while rhinos can reach several metric tons (Table 1). Medial angle is associated with the presence of a “knee locking” mechanism (Shockey, 2001; Schuurman, Kersten & Weijs, 2003; Janis et al., 2012). Largely studied through the anatomy of the distal femora, this mechanism has barely been considered on the patellar side (Kappelman, 1988; Hermanson & MacFadden, 1996; Janis et al., 2012; Etienne et al., 2020a). Our results do confirm the high correspondence between the presence of a developed medial angle on the patella and that of a developed medial trochlear ridge on the femur (Hermanson & MacFadden, 1996; Mallet et al., 2019). This developed medial angle shared between equids and rhinos therefore appears as convergent in the two clades, implying that this structure may not be totally evolutionary and developmentally homologous in these two clades. Indeed, while the medial angle in equids is extended by a parapatellar cartilage on which the medial patellar ligament attaches (Hermanson & MacFadden, 1996: fig. 2), this cartilage seems absent in rhinoceroses (or is little developed–C. Etienne, personal communication), where the bony angle is more extended medially than in equids. As shown in our results, the medial angle in rhinos is more extended and ossified medially compared to equids. This could be directly related to their higher body mass, making cartilage alone insufficient to support the weight of the body and ensure an efficient “knee locking”. This would be in line with our second hypothesis, assuming different shapes of patella to ensure an efficient “knee locking” mechanism between equids and rhinos.

Similarly, the craniocaudal development of the base in equids offers a larger surface area for insertions of knee extensors, these muscles being more powerful in equids than in rhinos compared to the respective size of these animals (Etienne, Houssaye & Hutchinson, 2021). Myological data lack for tapirs but the reduced surface of the base likely indicates reduced muscle insertions and, therefore, less powerful knee extensors in these animals. Such a craniocaudal development shifts the insertion of the extensor muscles away from the axis of rotation of the knee, thus increasing the moment arm of the joint and the muscular efficiency. This craniocaudal development of the patella in equids likely underlines the extreme cursorial specialization of this clade, while the condition in rhinos appears more as a trade-off between the support of their high body mass, their ability to reach a relatively high speed, and their evolutionary legacy.

The convergent emergence of the knee asymmetry in these clades remain barely understood among ungulates, various authors having supposed a link with either locomotor habit, feeding behaviour or body mass (Kappelman, 1988; Hermanson & MacFadden, 1996; Shockey, 2001; Shockey et al., 2008; Danaher, Shockey & Mihlbachler, 2009; Janis et al., 2012; Mihlbachler et al., 2014; Etienne et al., 2020a). Locomotor habit shows a marked dichotomy between these clades. Rhinos and equids both practice a transverse gallop while tapirs practice rotatory gallop, involving different limb loadings during the gallop phases (Economou et al., 2020). Rotatory gallop involves a phase where both hindlimbs hit the ground, while this never happen in transverse gallop, leading to higher mechanical stress exerted into a single limb in the latter gait. While equids present an extreme cursorial specialization associated with relatively low mass (Carrano, 1999), weight-bearing rhinos are still able to reach high speed despite weighting several tons (Alexander & Pond, 1992; Etienne, Houssaye & Hutchinson, 2021). An asymmetrical patella could be related to the emergence of a similar gallop in both equids and rhinos, compared to tapirs. Differences in body mass may then explain the further divergences existing in patellar shape between these two clades practicing transverse gallop. Further investigations including a large set of fossil perissodactyls are needed to sharpen the understanding of this puzzling feature and the functional advantages of this presence or absence in ungulates.

Conclusion

Our investigation of the shape of the patella among modern rhinoceroses and related extant perissodactyls reveals a higher interspecific than intraspecific morphological variation. Contrary to our predictions, and despite its central functional role in the knee joint and its implication in locomotion, the patella is little affected by variation of body mass between and within species. Rather, the patellar shape follows clear morphotypes existing in each species and even more marked at the scale of the family. This strong shape conservatism may be directly related to its developmental origin, shared with the femur, another bone whose shape has been proved to be highly linked to phylogenetic affinities before functional constraints in perissodactyls. Despite a strong evolutionary legacy leading to a relative shape inertia, the shared presence of a medial angle in rhinos and equids constitutes a clear case of morphological convergence highlighting that functional constraints are also at work on the shape of this sesamoid bone. This asymmetric conformation of the patella is related to a “knee locking” mechanism and maybe to the shared emergence of transverse gallop in both clades. Within this convergence, rhinos present additional shape modifications that may be related to the support of a heavy mass. Further investigations of the shape of the patella among a larger set of extinct Perissodactyla may be helpful for better understanding the morphofunctional evolution of this sesamoid bone in mammals.

Supplemental Information

Supplemental Information 1 PCA plot of the result of repeatability test.

Each anatomical landmark configuration was digitized ten times on three specimens of Dicerorhinus sumatrensis chosen to display the fewer morphological difference as possible. Each colour corresponds to a specimen (green: AMNH M-81892, black: NHMUK ZD 1879.6.14.2, red: NHMUK ZE 1948.12.20.1). The inter-specimen variation is lower than the intra-specimen error due to differences between landmark digitization. We concluded to the relevance of our anatomical landmark configuration to describe shape variation within our sample.

Supplemental Information 2 Results of the Procrustes ANOVA on shape data against log-transformed femoral circumference for rhinos.

A: Regression plot with theoretical shapes associated with minimum and maximum fitted values (respectively in medial and caudal views). Colour code follows Figure 2. Point size is proportional to the log centroid size of each specimen. Symbols indicate the sex attribution (triangle: male; circle: female; square: unknown). Silhouettes of C. simum, Dc. bicornis, Ds. sumatrensis, R. sondaicus, and R. unicornis are personal creations. All other silhouettes provided by www.phylopic.org under the Creative Commons license. Tree generated by our R code provided as Supplemental Data. B: Colour maps of the location and intensity of the shape deformation. The shape associated with the maximal femoral circumference value of the Procrustes ANOVA was coloured depending on its distance to the shape associated with the minimal value. Green indicates no deformation; blue indicates a negative deformation of high intensity; red indicates a positive deformation of high intensity. Plot and theoretical 3D models generated by our R code provided as Supplemental Data (using the specimen Dicerorhinus sumatrensis AMNH M-81892 as a template for deformation of the meshes).

Supplemental Information 3 Shape variation of the patella within the family Tapiridae.

A: Neighbour Joining tree computed on all PC scores obtained from the PCA performed on shape data of tapirs only. Colour code follows Figure 2. Symbols indicate age class as in Table 1 (triangle: subadult; circle: adult). Point size is proportional to the log centroid size of each specimen. Silhouette of T. indicus is personal creation. All other silhouettes provided by www.phylopic.org under the Creative Commons license. Tree generated by our R code provided as Supplemental Data. B: Morphospace of the two first axes of the PCA performed on morphometric data of tapirs and minimal and maximal theoretical shape associated with this variation (respectively in medial and caudal views). Colour codes follow Figure 2. Symbols indicate age class as in Table 1 (triangle: subadult; circle: adult). Point size is proportional to the mean log centroid size of each specimen. T. indicus, the heaviest tapir and sister-group of all other three species, occupies highest values on PC1 and lowest on PC1. T. bairdii occupies null values for both axes. T. terrestris and T. pinchaque, the lightest species being sister-taxa together, occupy lowest PC1 values and highest PC2 values. Plot and theoretical 3D models generated by our R code provided as Supplemental Data (using the specimen Tapirus terrestris RBINS 1185D as a template for deformation of the meshes).

Supplemental Information 4 Shape variation of the patella within the family Equidae.

A: Neighbour Joining tree computed on all PC scores obtained from the PCA performed on shape data of equids only. Colour code follows Figure 2. Symbols indicate age class as in Table 1 (triangle: subadult; circle: adult). Point size is proportional to the log centroid size of each specimen. Silhouettes of E. z. hartmannae and E. grevyi are personal creations. All other silhouettes provided by www.phylopic.org under the Creative Commons license. Tree generated by our R code provided as Supplemental Data. B: Morphospace of the two first axes of the PCA performed on morphometric data of equids and minimal and maximal theoretical shape associated with this variation (respectively in medial and caudal views). Colour codes follow Figure 2. Symbols indicate age class as in Table 1 (triangle: subadult; circle: adult). Point size is proportional to the mean log centroid size of each specimen. Most zebras occupy highest PC1 and PC2 values but remain mixed with some donkeys and horses. E. hemionus is isolated towards the lowest PC2 values. Donkeys and horses occupy mostly null and negative PC1 values. Plot and theoretical 3D models generated by our R code provided as Supplemental Data (using the specimen Equus quagga chapmani RBINS 1218 as a template for deformation of the meshes).

Supplemental Information 5 Patellar mean shapes for each species of the dataset (medial and caudal views respectively).

Shapes are computed as the mean conformations of all specimens per species. For species represented by a single specimen, shapes correspond to the specimen.Colour code follows Figure 2. Silhouettes of C. simum, Dc. bicornis, Ds. sumatrensis, E. z. hartmannae, E. grevyi, R. sondaicus, R. unicornis and T. indicus are personal creations. All other silhouettes provided by www.phylopic.org under the Creative Commons license. Theoretical 3D models generated by our R code provided as Supplemental Data (using the specimen Diceros bicornis NHMUK ZD 1879.9.26.6 as a template for deformation of the meshes).

Supplemental Information 6 Results of the Procrustes ANOVA on shape data against log-transformed femoral circumference for all perissodactyls.

A: Regression plot with theoretical shapes associated with minimum and maximum fitted values (respectively in medial and caudal views). Colour code follows Figure 2. Point size is proportional to the log centroid size of each specimen. Silhouettes of C. simum, Dc. bicornis, Ds. sumatrensis, E. z. hartmannae, E. grevyi, R. sondaicus, R. unicornis and T. indicus are personal creations. All other silhouettes provided by www.phylopic.org under the Creative Commons license. B: Colour maps of the location and intensity of the shape deformation. The shape associated with the maximal femoral circumference value of the Procrustes ANOVA was coloured depending on its distance to the shape associated with the minimal value. Green indicates no deformation; blue indicates a negative deformation of high intensity; red indicates a positive deformation of high intensity. Plot and theoretical 3D models generated by our R code provided as Supplemental Data (using the specimen Diceros bicornis NHMUK ZD 1879.9.26.6 as a template for deformation of the meshes).

Supplemental Information 7 Detailed list of the studied specimens with MorphoSource ARK Identifier for each 3D model.

Supplemental Information 8 R code used for the analyses.

Supplemental Information 9 Landmark conformations gathered on the 3D models.

For each specimen, the dataset is composed of the coordinates of the 6 anatomical landmarks and 103 curve sliding semi-landmarks. The template (C. simum AMNH M-51854) is composed in addition of 461 surface sliding semi-landmarks.

The authors warmly thank all the curators of the visited institutions for granting us access to the studied specimens: E. Hoeger and S. Ketelsen (American Museum of Natural History, New York, USA), C. West, R. Jennings, M. Cobb (Powell Cotton Museum, Birchington-on-Sea, UK), J. Lesur, A. Verguin (Muséum National d’Histoire Naturelle, Paris, France), R. Portela-Miguez (Natural History Museum, London, UK), F. Zachos, A. Bibl (Naturhistorisches Museum Wien, Vienna, Austria), O. Pauwels, S. Bruaux and A. Folie (Royal Belgian Institute of Natural Sciences, Brussels, Belgium), E. Gilissen (Royal Museum for Central Africa, Tervuren, Belgium) and A. H. van Heteren (Zoologische Staatssammlung München, Munich, Germany). We acknowledge the EDDyLab (ULiège, Belgium) and V. Fischer for allowing us to use the Creaform HandySCAN 300 laser surface scanner to scan several specimens. The authors also warmly thank L. Moizo (MNHN, Paris, France), who reconstructed and extracted 3D models during his MSc internship. We acknowledge C. Étienne (MNHN, Paris, France) for precious indications regarding the knee anatomy in rhinos, and J. Gônet (MNHN, Paris, France) for digitizing some additional patellas. We would like to thank J. Wölfer (Humboldt-Universität zu Berlin, Germany) and an anonymous reviewer for their precious comments that improved the quality of the manuscript.

Additional Information and Declarations

Competing Interests

Author Contributions

Data Availability

The authors declare that they have no competing interests.

Christophe Mallet conceived and designed the experiments, performed the experiments, analyzed the data, prepared figures and/or tables, authored or reviewed drafts of the article, and approved the final draft.

Alexandra Houssaye conceived and designed the experiments, analyzed the data, authored or reviewed drafts of the article, and approved the final draft.

The following information was supplied regarding data availability:

The R code used to performed the 3D geometric morphometrics are available in the Supplemental File.

The 54 3-D models are available at MorphoSource: https://www.morphosource.org/projects/000366286?locale=en.

The data related to the Institute of Natural Sciences (Belgium) is permanently public, and users can view the media record, but the curators require interested parties to request access to download the data in order to measure the uses made of this investment, prevent any fraudulent or commercial abuses, and allow the curators to measure how these collections are being used, in order to demonstrate how useful they are to the public and the scientific community, and provide good arguments for further investment in data digitization.

Please contact the curators for download approval: Annelise Folie, Curator of Paleontology Collections, afolie@naturalsciences.be; Olivier S. G. Pauwels, Curator of Recent Vertebrate Collections, opauwels@naturalsciences.be.

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
