# Peer review of "Deciphering the influence of evolutionary legacy and functional constraints on the patella: an example in modern rhinoceroses amongst perissodactyls"

_PeerJ, doi:10.7717/peerj.18067_

## Round 0.1 · original submission · Major Revisions

Both reviewers have provided mostly positive feedback, despite recommending a significant number of changes. Reviewer 1 highlights the need to correct the misleading title about "functional constraints," addresses the problem of inadequate sampling which undermines the support for certain findings, raises questions about the disproportionate focus on rhinoceroses, points out the disorganized nature of the results section, and suggests a clearer, comparative analysis of patellar morphology. Additionally, this reviewer advises on refining the presentation of quantitative findings and enhancing the clarity of species labels in figures, which would bolster the manuscript’s contribution to the study of perissodactyl morphology and evolution.

Reviewer 2 notes the study's broad scope, blending exploratory analysis with hypothesis testing, potentially confusing the reader about its aims. They advocate for a more methodical approach to hypothesis testing, such as investigating the knee locking mechanism's impact on talar shape, and call for an improvement in describing statistical methods to focus the paper more sharply. The absence of comprehensive statistical results in the main document and supplementary materials is criticized, highlighting the necessity for a better display of these findings to elucidate their impact on shape analysis more clearly. While the exploratory results are seen as valid, the inferential statistical analysis is deemed in need of refinement for accurate interpretation. Specific enhancements are suggested, including how statistical models could more effectively demonstrate changes in shape, to clarify the study’s results before reaching conclusions.

I expect all these points, as well as others raised by both reviewers, to be addressed.

Reviewer 1 ·

Basic reporting

see below

Experimental design

see below

Validity of the findings

see below

Additional comments

This paper is an interesting morphometric analysis of phenotypic variation in the patellae of extant Perissodactyla. I am satisfied that the results and conclusions about the three hypotheses presented in the paper are justified by the analysis that is presented. However, the paper is not without need for improvement. I have provided a few major comments below, with many minor comments and other suggestions for editorial change inserted within the manuscript. Once improved , I believe this paper will make an excellent contribution to perissodactyl morphology and will pave the way for studies that include fossil taxa to further investigate questions about perissodactyl evolution.

Major comments
1. The title is somewhat misleading. The paper reports on shape differences and investigates the influence of phylogeny and body size on patellar morphology. To say the paper addresses "functional constraints" is a stretch. I suggest removing it from the title.

2. While the three main hypotheses are adequately addressed, some of the more specific observations provided in the paper are not supported due to insufficient sampling. This includes statements about the distribution of juvenile patellar morphology, and the degree to how certain species are distributed in the various dendrograms and bivariate plots. Some species are represented by a very small number of specimens and there are very few juveniles in the sample overall. There is not sufficient data to make robust observations on the distribution of variation in these subsamples.

3. Why are rhinoceroses investigated more intensely in the main paper and other family-specific analysis are relegated to the supplemental materials?

4. The results section is presently an extended discussion of quantitative results mixed with descriptions of patellar morphology. This section seems to lack direction and coherency and is very tedious. I think it would be helpful if a comparative description of perissodactyl patellar morphology were provided separately from the morphometric results. Presently, the paper does not explicitly recognize that the shape differences between rhino, tapir, and horse patellae are plainly visible and have already been abundantly documented without quantitative analyses. Accompanying this comparative description should be a figure that readily shows rhino, horse, and tapir patellae side by side so that the reader can see the readily observable differences. At present there is no figure in the paper that makes the differences plainly visible. The description should be comparative. Rather than describing patellar features in isolation (as is abundantly done in the present manuscript), please describe the patellae comparatively so that the reader can easily understand the differences.

5. Once a main understanding of basic comparative patellar morphology has been established, the quantitative results can be reported much more succinctly and in ways that more directly address the hypotheses of the paper. In revising this section, the accuracy of how the various dendrograms and bivariate are described needs improvement. Eliminate comments that about subsets of the data for which there are not enough observations to make robust statements about their distributions.

6. Finally, I would appreciate revision of the species labels in the figures, particularly ones such as Figure 10, where many species are figured at once. Many of the animal symbols look very similar and some of the colors (particullary red) are used for more than one species.

Annotated reviews are not available for download in order to protect the identity of reviewers who chose to remain anonymous.

·

Basic reporting

The manuscript is overall really well-written and easy to access and the figures are high quality. The reported tables are, however, not sufficient and lack some important results. The literature is well-referenced and the study is put into a clear context. This study is partly exploratative, but also contains hypotheses, which makes it difficult to grasp the scope of the study. Some of these hypotheses are tested statistically, while others are not without a clear explanation why.

Experimental design

The research questions are in part well-defined, howerver, this study is also very exploratative as explained by the authors. The research questions are meaningful and relevant and the authors made an effort to rigorously analyse their dataset. However, the statistical methods are insufficiently described and leave room for improvement. I think that the focus of the paper (despite its exploratative character) can be much more sharpend by testing all hypothesis in a systematic manner (e.g., the effect of the knee locking mechanism on the talar shape). In the attached PDF, I explained how the regression/anova analyses could be condensed and improved with regard to the many factors that are analysed in their effect on talar shape. Also, not all statistical results were provided in table or figure format (also not in the supplementary materials, as far as I know), which I consider important.

Validity of the findings

I consider the results of the exploratative aspect of this study valid, such as the PCA analyses and the neighbor joining analyses. However, the inferential statistical anaylsis need some improvement for proper interpretability. I think that especially how the statistical models are used to translate their regression trends into shape illustrations is important to get a grasp on what the models actually explain about shape (see attached PDF for recommandations). Before these aspects are not clarified, I think it is not possible to make a final statement about the discussion and conclusion (except for it being well-written).

---

## Round 0.2 · Minor Revisions

Dear authors, we are very close to finishing the process. Please incorporate the suggested revisions in this final round. Thank you very much.

In the references, change Corina Miriam V. to Vera MC.

Reviewer 1 ·

Basic reporting

see comments below

Experimental design

see comments below

Validity of the findings

see comments below

Additional comments

I am satisfied with how the authors addressed my concerns. This is a nice study.

·

Basic reporting

The authors did a very good job adressing my previous concerns. The analysis was simplified and I think that the manuscript much benefitted from this simplication (removing age and interactions effects from the analysis).

Experimental design

Everything looks very sound. I just added some minor comments to the M&M section.

Validity of the findings

The results are well-presented and discussed. I added a minor comment on the discussion about how size and phylogenetic affinity cannot be perfectly separated, which I would like to see incorporated into the discussion.

Additional comments

The authors improved on the ms and the whole study is very coherent, now. The figures and tables are high quality and make it easy to follow along the main text. I am looking forward to seeing this study published.

---

## Round 0.3 · accepted · Accept

Thank you very much for the attention the authors have given to the detailed and rigorous suggestions from our reviewers. We are ready to move forward; congratulations on a very interesting work.